# Fractional Ablative Carbon Dioxide Lasers for the Treatment of Morphea: A Case Series and Literature Review

**DOI:** 10.3390/ijerph19138133

**Published:** 2022-07-02

**Authors:** Paulina Klimek, Waldemar Placek, Agnieszka Owczarczyk-Saczonek

**Affiliations:** Department of Dermatology, Sexually Transmitted Diseases and Clinical Immunology, University of Warmia and Mazury in Olsztyn, Oczapowskiego 2, 10-719 Olsztyn, Poland; w.placek@wp.pl (W.P.); aganek@wp.pl (A.O.-S.)

**Keywords:** morphea, localised scleroderma, fractional ablative carbon dioxide laser, FAL

## Abstract

Morphea is an inflammatory, immune-mediated disease of unknown aetiology. It is characterised by excessive collagen deposition, which leads to the hardening of the dermis and subcutaneous tissues. The disease is associated with cosmetic and functional impairment, which can affect the patients’ quality of life. Fractional ablative lasers (FALs) are currently used for the treatment of many skin diseases that are connected to tissue fibrosis due to the low risk of side effects and their great effectiveness. This study aimed to improve the aesthetic defects that are caused by morphea lesions and assess the efficacy and safety of FAL use in this indication. We also reviewed the literature on the subject. We present four women with biopsy-proven morphea, manifesting as hyperpigmented plaques and patches. One of the patients additionally had morphea-related knee joint contracture. Four fractional CO_2_ laser sessions, separated by one-month intervals, were performed and produced significant improvements in dyspigmentation and induration. An improved elasticity and a decrease in dermal thickness were also obtained, as proven by measurements using DermaLab Combo. No severe adverse effects occurred. Based on these cases presented by the authors, fractional CO_2_ lasers appear to be an effective, well-tolerated, and safe therapeutic option for patients suffering from morphea.

## 1. Introduction

Morphea (localised scleroderma, LoSc) is a chronic inflammatory fibrosing disorder. The disease mainly affects the skin but can also involve deeper structures, such as fat, fascia, muscle, and bone [1]. The incidence of LoSc ranges from 0.4 to 2.7 per 100,000 people per year, with two peaks at the age of onset: between 7 and 11 years of age in children and between 40 and 50 years of age in adults. It is more common in women and in the Caucasian population [2,3].

The aetiopathogenesis of LoSc is still not clearly understood. It involves genetic, epigenetic, and environmental factors, such as infections, trauma, radiation or drugs, together with immune and vascular disorders. These factors may activate keratinocytes to release inflammatory mediators, which stimulate lymphocytes, endothelial cells, and fibroblasts to increase the synthesis of collagen I and III [3,4,5,6]. Transforming growth factor-beta (TGF-B) plays an important role in this process. It stimulates fibroblasts for the production of collagen, fibronectin, and proteoglycans, while at the same time inhibiting the synthesis of matrix metalloproteinases (MMP) (which are responsible for collagen degradation) [7,8].

There are three phases of the disease: early inflammatory (active), progressive sclerosis, and atrophy. Initially, lesions are erythematous and slightly infiltrated, sometimes with pain and itching. In some cases, the main symptom from the onset of the disease is hyperpigmentation plaques without the presence of sclerosis. The progressive lesions can change colour. An area of increased cohesiveness is a porcelain-white colour with an inflammatory border (*lilac ring*), which is a marker of an active inflammatory process. The inactive period can include brown discolouration and features of skin atrophy, including a shiny surface, hypopigmentation, translucent blood vessels, and the destruction of hair follicles [1,3,4]. 

Morphea classification systems are based on clinical phenotypes. The authors of the German guidelines proposed a classification that takes into account the severity, extent, and depth of the fibrotic process, thus differentiating five main clinical variants: limited, generalised, linear, deep, and mixed [1]. 

The choice of therapy is influenced by the clinical type of the localised scleroderma, the activity of the disease, the extent and depth of the lesions, and the features of the tissue damage. In general, LoSc treatment options include topical and systemic therapies, as well as ultraviolet (UV) phototherapy [8]. 

Morphea is associated with cosmetic, functional, and psychosocial impairment, which can affect the patients’ quality of life [9]. Some lesions may remit spontaneously, but limb contractures and aesthetic defects, such as atrophy, hyperpigmentation, and hypopigmentation, are unlikely to regress. Skin lesions are often localised on exposed areas of the body (face, limbs) [10], which leads to stigmatisation. 

As proposed by consensus, the therapeutic options tend to inhibit disease progression but do not include methods for the aesthetic correction of the disfigurement that is caused by the disease [1,3,8]. Therefore, it is clear that a search for safe and effective practical methods to correct these cosmetic and functional defects is desirable [11].

The literature provides reports on the correction of aesthetic defects using laser and light-based therapy, injectables, and surgical interventions [12]. Promising results from treatment with a fractional ablative carbon dioxide laser [13,14,15,16], pulsed dye laser (PDL) [17,18,19,20,21,22,23,24], excimer laser [25,26,27,28], Er:YAG fractional laser [29,30], Q-switched alexandrite laser [31], and long-pulsed Nd:YAG laser associated with pulsed light [32] have been presented.

The subject of the correction of an aesthetic defect that was caused by localised scleroderma remains within the scope of the authors’ interests. We have proven that hyaluronic acid injections can be an effective and safe therapeutic option for patients suffering from LoSc, according to the example of three women with lesions on the facial area. One of the patients also had fractional ablative CO_2_ laser (FAL) therapy to improve their skin hyperpigmentation [33].

The aim of this study was to improve the aesthetic defects that are caused by morphea lesions and assess the efficacy and safety of the use of fractional ablative CO_2_ lasers for the treatment of the disease. This article presents our experience on the subject. In this paper, we present a case series of four female patients with LoSc. 

## 2. Materials and Methods

### 2.1. Patients 

Four patients who were being treated at the Clinic of Dermatology, Sexually Transmitted Diseases, and Clinical Immunology in Olsztyn for histopathologically proven morphea were included in the study. The active phase of the disease was ruled out. The patients provided informed consent to participate in the study and for the publication of the results. All of the patients had contraindications to fractional laser therapy excluded. 

### 2.2. Clinical Evaluation 

The results were described using the Localised Scleroderma Damage Index (LoSDI), which is part of the Localised Scleroderma Assessment Tool (LoSCAT). The LoSDI assesses dermal atrophy (DAT), subcutaneous atrophy (SAT), and dyspigmentation (DP). Scores range from 0 to 3 in each domain, with 9 points in total indicating the most severe skin damage. 

The LoSDI scores were obtained both before and one month after FAL therapy. 

The patients were examined every four weeks without observing any adverse reactions. The observation time ranged from two months to two years. 

### 2.3. Ultrasound Assessment of Dermal Thickness 

Dermal thickness measurements were assessed by measuring the biophysical parameters of the skin using the Ultrasound DermaLab Combo 01/2014CLINIQUE, Cortex Technology, Hadsund, Denmark. Images were obtained by manipulating a B-mode high-resolution ultrasound system using a 20-MHz centre frequency transducer. The transducer was positioned perpendicular to the longitudinal axis of the skin in the centre of the assessed skin lesion. The thickness of the dermal skin was calculated in micrometres (μm), based on a super A scan. 

Ultrasonography images were obtained both before and one month after FAL therapy.

### 2.4. Elasticity Assessment 

Elasticity measurements were assessed by measuring the biophysical parameters of the skin using the Ultrasound DermaLab Combo 01/2014CLINIQUE, Cortex Technology, Hadsund, Denmark. 

The elasticity measurements from the DermaLab Combo are based on suction that is applied to the skin surface. The retraction time (R) is the time in milliseconds (ms) that it takes for the skin to retract from the peak elevation to 33% of the peak elevation. 

Retraction times were measured both before and one month after FAL therapy. 

### 2.5. Fractional Ablative Carbon Dioxide Laser Treatment 

The FAL procedure was applied monthly (four to eight times in total) on selected skin lesions using a fractional CO_2_ laser (Metrum Cryoflex, Blizne Łaszczyńskiego, Warsaw, Poland). 

The treatment parameters were as follows: power = 40 W, microbeam spacing = 1200 μm, and energy density = 40 mJ/cm^2^ or power = 30 W, microbeam spacing = 1400 μm, and energy density = 36 mJ/cm^2^.

The treatment sites were anesthetised with a cream containing 5% lidocaine and 5% prilocaine thirty minutes before the procedure.

## 3. Case Series

### 3.1. Case 1

A thirty-year-old female who had had biopsy-proven linear scleroderma of the extremities since she was eight years old was admitted to the dermatology department to modify her disease treatment. After the LoSC was diagnosed, she was initially treated with methotrexate for five years, then with topically applied glucocorticoids, physiotherapy, and manual therapy. Recently, she had been taking pentoxifylline and chloroquine.

A physical examination revealed longitudinally, irregularly hyperpigmented, and indurated lesions on the posterior medial region of the left lower limb and in the left inguinal area. The patient reported a restriction of motion in the left knee joint. The laboratory results and imaging did not show any significant abnormalities. No contraindications for continuing chloroquine were found during the ophthalmological consultation.

During her hospitalisation, the first course of laser treatment on the left leg was performed (power = 40 W, microbeam spacing = 1200 μm, and energy density = 40 mJ/cm^2^) with good tolerance. The patient was discharged with the recommendation to continue FAL therapy and pentoxifylline with chloroquine.

After four monthly laser treatments on the knee and calf, the LoSDI score was lowered from 7 (DAT 3, SAT 2, and DP 2) to 3 (DAT 1, SAT 1, and DP 1) (Figure 1a,b). The patient reported a subjective improvement in her range of motion. We decided to broaden the scope of the inguinal lesions and also obtained a good aesthetic result, which was reflected by the LoSDI score of 4 (DAT 1, SAT 2, and DP 1) compared to the LoSDI score of 8 (DAT 3, SAT 2, and DP 3) before the first procedure (Figure 1c,d). We used the lower parameters of the device (power = 30 W, microbeam spacing = 1400 μm, and energy density = 36 mJ/cm^2^) in the hip area and performed four sessions with the same time intervals.

### 3.2. Case 2

A thirty-one-year-old woman was admitted to the dermatology department with a one-year history of non-indurated brown patches on her forearms. She denied the presence of erythematous lesions or *lilac rings*. The patient had no history of chronic diseases and no family history of autoimmune disorders.

On physical examination, symmetrically hyperpigmented plaques without induration were present on distal parts of the upper limbs. The initial LoSDI score was 2 (DAT 0, SAT 0, and DP 2). The laboratory tests showed no abnormalities (ANA-HEP2 titer 1:160). Atrophoderma of Pasini and Pierini (superficial morphea) was diagnosed, based on the clinical presentation and a histopathological examination.

Due to the lack of features of disease activity, we decided to start fractional ablative carbon dioxide laser therapy to improve the cosmetic effects that had been caused by the disease. The procedure was applied monthly (four times in total) on every skin lesion (power = 40 W, microbeam spacing = 1200 μm, and energy density = 40 mJ/cm^2^). One month after the last course of laser treatment, we observed a great improvement in the skin discolouration (a LoSDI score of 0: DAT 0, SAT 0, and DP 0) (Figure 2a,b). The patient was satisfied with the effects of the therapy. During a follow-up visit six months later, there was no need to repeat the treatment because the good effects had persisted. No adverse effects were observed. 

### 3.3. Case 3

In this fifty-one-year-old patient, we observed lesions on her extremities that were oval-shaped, indurated, and brownish with an ivory colour in the centre of the lesions and many round-shaped hyperpigmented lesions on her abdomen. She had been under our care since she was diagnosed with plaque morphea on the basis of histopathology and a clinical picture from five years earlier.

The patient also suffered from Hashimoto’s disease and a small intestinal bacterial overgrowth. The treatment of her LoSc involved procaine penicillin, bath PUVA therapy, and subcutaneous injections of triamcinolone.

After the stabilisation of the disease, the aesthetic defects remained, which significantly influenced the patient’s well-being. Indurated lesions on her shanks received four laser sessions, separated by one-moth intervals, using a fractional CO_2_ laser (power = 40 W, microbeam spacing = 1200 μm, and energy density = 40 mJ/cm^2^). The LoSDI score was lowered from 4 (DAT 1, SAT 0, and DP 3) to 2 (DAT 1, SAT 0, and DP 1) (Figure 3a,b).

Six months after the last procedure, we decided to repeat the laser treatment to achieve a better cosmetic effect. We performed four more laser courses with the same treatment parameters. This resulted in the lowering of the LoSDI score to 1 (DAT 0, SAT 0, and DP 1) (Figure 3c).

### 3.4. Case 4

A sixty-two-year-old female with a three-month history of brown patches on her trunk, which was histopathologically described as localised scleroderma, was reported to the dermatology department. She further suffered from chronic urticaria and hypertension. 

Upon physical examination, irregular hyperpigmented non-indurated patches with a shiny surface were present on her abdomen. Subsequent additional evaluations, including morphology and biochemistry, ANA-HEP2, IgM and IgG antibodies against *B. burgdorferi*, a chest X-ray, and physical urticaria tests, were negative. 

The skin patch tests were positive for rosin, Peruvian balsam, propolis, and linalool. During hospitalisation, the patient started PUVA phototherapy, which she continued after discharge. 

One year later, the patient underwent four FAL sessions for the morphea lesions on her abdomen. A very good aesthetic effect was obtained, which was reflected by a LoSDI score of 0 (DAT 0, SAT 0, and DP 0) compared to the LoSDI score of 2 (DAT 0, SAT 0, and DP2) from before the laser therapy (Figure 4a,b).

The cases presented in this article are summarised in Table 1.

## 4. Results

### 4.1. Clinical Evaluation 

In all of the patients, a significant improvement in dermal atrophy, subcutaneous atrophy, and dyspigmentation was achieved, which was illustrated by the decreased LoSDI scores for each assessed skin lesion.

Only mild, transient adverse effects, such as erythema, oedema or pain during the procedure, were observed. No severe adverse effects occurred. 

### 4.2. Ultrasound Assessment of Dermal Thickness 

After the laser therapy, we obtained a decrease in the dermal thickness of each assessed morphea lesion (Table 2).

### 4.3. Elasticity Assessment 

In the post-treatment measurements, retraction time (ms) was decreased in every patient, which showed an improvement in elasticity (Table 3).

## 5. Discussion

The current standards of morphea treatment focus on the inhibition of the progression of the disease; however, reducing the aesthetic defects that are caused by the disease seems to be equally important for patients. 

The development of FAL technology has allowed us to effectively correct aesthetic defects, such as scars, hyperpigmentation, and superficial or deep rhytids [34]. It also offers novel treatment for morphea and other connective tissue diseases, which often cause permanent disfigurement [35].

A MEDLINE literature search using the terms *localised scleroderma*, *morphea*, *fractional laser*, and *carbon dioxide laser* showed five case reports or case series, which presented twenty-two patients who had been treated with a fractional ablative carbon dioxide laser for morphea. 

In 2011, Kineston et al. (for the first time in the literature) described a case of the use of FAL for a patient with localised scleroderma. The authors observed significant improvements in the range of motion for a twenty-seven-year-old woman with morphea-related contracture. The patient underwent one laser session on the indurated parts of her lower limb via a fractional ablative 10.6-μm carbon dioxide laser (Ultrapulse Encore Deep FX; Lumenis, Ltd., Santa Clara, CA, USA) using the following settings: single pass, single pulse, no overlap, 50-mJ pulse energy, and 5% density [13]. 

Subsequently, Shalaby et al. compared the efficacy of the use of FAL for the treatment of morphea to the use of UVA-1 phototherapy. FAL showed a statistically significant advantage over UVA-1 according to the clinical lesion assessment (LoSCAT). In a histopathological assessment, improvements in collagen homogeneity and inflammatory penetration were observed, but the difference between the groups was not statistically significant. An immunohistochemical assessment revealed significantly decreased TGFb1 and increased matrix MMP1, as well as a decreased skin thickness in the ultrasound evaluation [14]. 

Farmer et al. reported a case series of two patients with linear morphea on the head or thighs, who were treated successfully with improvements in hyperpigmentation, induration, and range of motion using a fractional CO_2_ laser. Additionally, a patient with a parieto-frontal lesion received one round of deep FAL, followed by 0.3 cc of topical poly-L-lactic acid and botulinum toxin A [15]. A combination of topical and injected poly-L-lactic acid with FAL for the correction of deep morphea on the shoulder was also applied by Yeager with very good results [16]. 

The authors of this paper reported the case of a 70-year-old woman with linear morphea on the chin, who was treated effectively with hyaluronic acid injections and three monthly fractional laser sessions. The effects were confirmed by the patient and a clinical assessment, which included a reduction in the LoSDI score from 5 to 1. From our point of view, the combination of a hyaluronic acid fillers and FAL therapy allowed us to achieve better results than using one method only [33]. 

FAL is based on fractional photothermolysis. The device produces an array of point-heat damage zones on the surface of the skin (microthermal treatment zones, MTZs), which are surrounded by a margin of undamaged tissue [14,36]. This leads to an immediate mechanical effect that removes fibrotic tissue and releases skin tightness [37]. Near the MTZs, controlled collateral dermal heating is achieved. The controlled thermal stress is followed by a wound-healing response, ultimately leading to long-term dermal remodelling, thereby decreasing the level of TGFb and causing an induction of the expression of metalloproteinases (MMP-1, MMP-3, MMP-9, and MMP-13) that are responsible for the degradation of improperly homogenised collagen fibres [35,36,37,38,39,40,41,42,43,44]. TGFb1 is a well-known fibrogenic factor that may play a role in the pathogenesis of morphea by disrupting the balance between collagen formation and degradation, thereby depositing extracellular matrices [2,4,6,7,42]. In response to the damage, there is also an increase in heat shock proteins (HSPs), specifically HSP-72 in the early stages of healing after laser therapy and HSP-47 for up to three months after the session. HSP-72 leads to the stimulation of stem epidermis cells and dermis initiation collagen remodelling, while HSP-47, which affects fibroblasts within and around MTZs, enables the long-term rebuilding of collagen. Gradually and irregularly, the arrangement of collagen bunches gives way to a new arrangement of orderly fibres. About a month after FAL treatment, type 3 collagen is replaced by type 1 [35,36,37,38,39,40,41,42]. 

One of the main aims of this study was to assess improvements in aesthetic defects that were caused by morphea lesions. The clinical assessment was described using the LoSDI. In all of the patients, a significant aesthetic effect was achieved. The treatment resulted in the lowering of the LoSDI score for each assessed skin lesion. In two out of the four patients, the skin lesions rated 0 on the LoSDI after the FAL therapy. In one of the women, six months after the last procedure, we performed four more laser courses to achieve a better cosmetic effect. The patient with knee-joint contracture reported subjective improvements in the range of motion in that joint.

The biophysical parameters were assessed using DermaLab Combo. All measurements were performed with probes for elasticity and ultrasound, in accordance with the manufacturer’s operating instructions [43]. High-frequency ultrasound was performed on all patients with a probe operating at 20 MHz, which allowed for the complete evaluation of the dermis and the upper subcutaneous tissues [44]. The DermaLab skin elasticity module uses a suction chamber to objectively evaluate the degree of fibrosis. Skin elasticity is determined by the refraction time (R) once the negative pressure is released [45]. A significant decrease in dermal thickness and an improvement in elasticity were noted compared to before the treatment for the entire group of patients. 

No severe adverse reactions were observed. The patients reported mild side effects, such as pain during the laser sessions or erythema and local oedema, which subsided in the first three days after the therapy. The proper use of a topical anaesthetic cream and appropriate post-treatment care recommendations helped to minimise these adverse effects. 

The use of laser treatments for autoimmune diseases, e.g., morphea, may be questionable and cause a fear of the recurrence of the disease [33]. Mechanical injuries are a known factor that can provoke the development of the disease [2]. For this reason, each invasive procedure, such as laser therapy, filler injections or surgical interventions, should be performed during the inactive phase of the disease [1,8]. Our patients’ observation times ranged from two months to two years; we did not observe any relapse of the inflammation stage of the disease, which aligned with previously cited cases. 

## 6. Conclusions

Morphea can lower patients’ quality of life due to the formation of permanent disfigurement and functional impairment. Safe methods for correcting aesthetic defects, such as dyspigmentation, induration, and skin atrophy, should be sought. 

Fractional ablative carbon dioxide laser therapy can be considered as a promising option for patients suffering from morphea. FAL therapy decreases dermal thickness, increases elasticity, and improves the clinical appearance of morphea lesions. It could be an effective, safe, and well-tolerated therapeutic method of treatment. The small group of patients was the limitation of this study. The method should be further explored in order to confirm its safety and efficacy on a larger experimental group.

## Figures and Tables

**Figure 1 ijerph-19-08133-f001:**
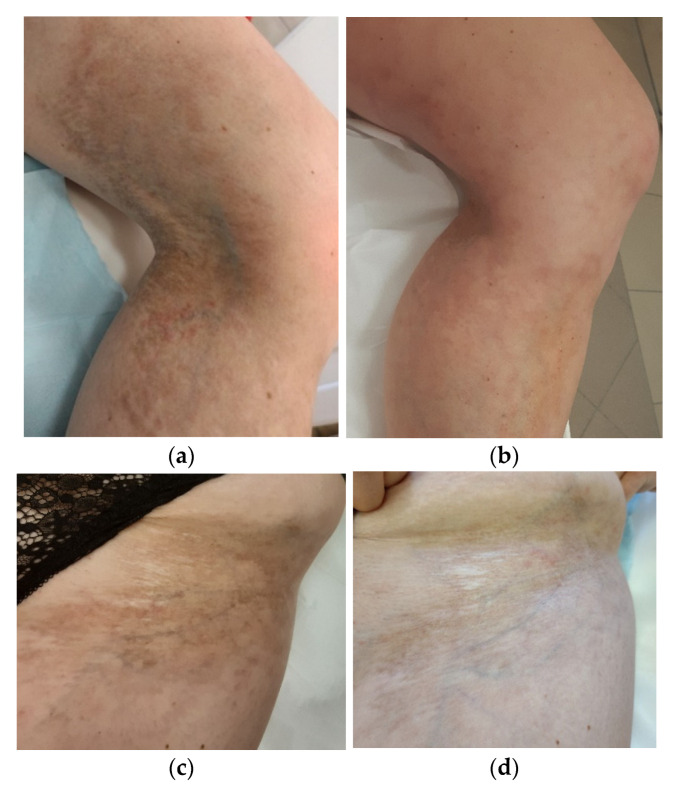
Patient no. 1: (**a**) before the FAL procedure with a LoSDI score of 7 (DAT 3, SAT 2, and DP 2); (**b**) six months after a series of four FAL therapies with a LoSDI score of 3 (DAT 1, SAT 1, and DP 1); (**c**) before the FAL procedure with a LoSDI score of 8 (DAT 3, SAT 2, and DP 3); (**d**) six months after a series of four FAL therapies with a LoSDI score of 4 (DAT 1, SAT 2, and DP 1).

**Figure 2 ijerph-19-08133-f002:**
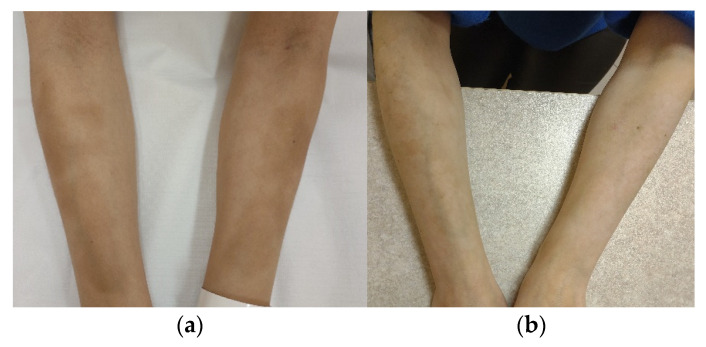
Patient no. 2: (**a**) before the FAL procedure with a LoSDI score of 2 (DAT 0, SAT 0, and DP 2); (**b**) six months after a series of four FAL therapies with a LoSDI score of 0 (DAT 0, SAT 0, and DP 0).

**Figure 3 ijerph-19-08133-f003:**
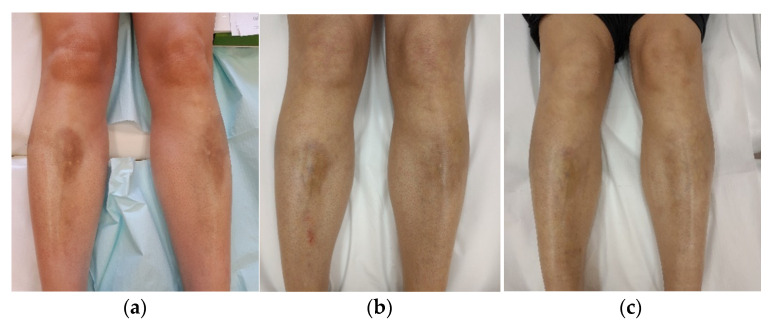
Patient no. 3: (**a**) before the FAL procedure with a LoSDI score of 4 (DAT 1, SAT 0, and DP 3); (**b**) six months after a series of four FAL therapies with a LoSDI score of 2 (DAT 1, SAT 0, and DP 1); (**c**) two months after a series of eight FAL therapies with a LoSDI score of 1 (DAT 0, SAT 0, and DP 1).

**Figure 4 ijerph-19-08133-f004:**
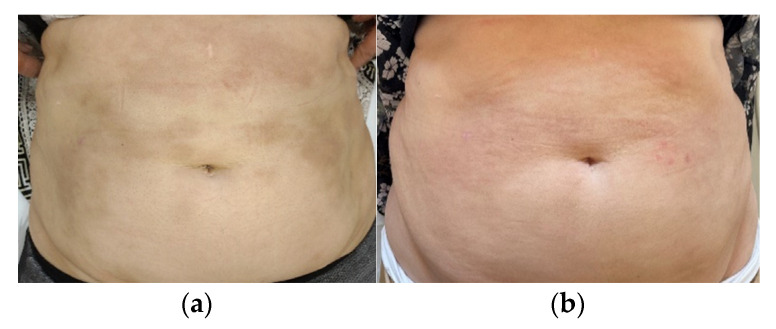
Patient no. 4: (**a**) before the FAL procedure with a LoSDI score of 2 (DAT 0, SAT 0, and DP 2); (**b**) two months after a series of four FAL therapies with a LoSDI score of 0 (DAT 0, SAT 0, and DP 0).

**Table 1 ijerph-19-08133-t001:** The case series.

	Age	Duration of Disease	Subtype of Morphea	Previous Treatment	No. of Treatments	Parameters	LoSDI before FAL	LoSDI after FAL
Case 1a *	30	22 years	Linear	Methotrexate, pentoxifylline, chloroquine, topically applied glucocorticoids, physiotherapy, and manual therapy	4	Power = 40 W and energy density = 40 mJ/cm^2^	7	3
Case 1b *					4	Power = 30 W and energy density = 36 mJ/cm^2^	8	4
Case 2	31	1 year	Atrophoderma of Pasini and Pierini	None	4	Power = 40 W and energy density = 40 mJ/cm^2^	2	0
Case 3	51	5 years	Plaque	Procaine penicillin, bath PUVA therapy, subcutaneous injections of triamcinolone, and topically applied glucocorticoids	8	Power = 40 W and energy density = 40 mJ/cm^2^	4	1
Case 4	62	1 year 3 months	Plaque	PUVA phototherapy and topically applied glucocorticoids	4	Power = 40 W and energy density = 40 mJ/cm^2^	2	0

* Case 1a, lesions on the knee area; Case 1b, lesions on the inguinal area.

**Table 2 ijerph-19-08133-t002:** Skin dermal thickness (μm).

	Baseline	4 Weeks after the 4th Procedure	Percentage Increment (%)
Case 1a *	976	882	−9.63
Case 1b *	1369	1080	−21.11
Case 2	1043	632	−39.4
Case 3	1435	1001	−30.24
Case 4	816	711	−12.87
			**−22.65**

* Case 1a, lesions on the knee area; Case 1b, lesions on the inguinal area.

**Table 3 ijerph-19-08133-t003:** Skin elasticity level (retraction time (ms)).

	Baseline	4 Weeks after the 4th Procedure	Percentage Increment (%)
Case 1a *	290	254	−12.41
Case 1b *	343	306	−10.78
Case 2	102	98	−3.92
Case 3	131	117	−10.68
Case 4	111	103	−7.2
			**−8.99**

* Case 1a, lesions on the knee area; Case 1b, lesions on the inguinal area.

## Data Availability

Not applicable.

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
