# Peer review of "Fractional Ablative Carbon Dioxide Lasers for the Treatment of Morphea: A Case Series and Literature Review"

_ijerph, 2022, doi:10.3390/ijerph19138133_

Round 1

Reviewer 1 Report

The article by Klimek et al prevents the results of fractional ablative carbon dioxide laser in their treatment of 4 women with morphea. The authors found improvement in the appearance of the lesions as shown by their photos of pre and post treatment, and in the scores of LoSDI. This is encouraging as there is a need for treatments to improve cosmetic appearance as the authors point out.

Major Criticisms
1. LoSDI scores unclear
a. The LoSDI is supposed to be used as a total score of skin damage signs, with the scores of its components summed across all affected anatomic sites, but the authors appear to be reporting scores for specific anatomic regions for case 1. Page 3, lines 133-134, they say that the LoSDI changed from 7 to 3. lines 135-137 discuss expanded laser treatment that includes inguinal lesion, with LoSDI changing from 8 to 4. Authors should clarify whether reported LoSDI are for specific regions and which regions they represent.
2. More clinical features of morphea should be presented.
a. The authors should discuss if induration is present in Case 3 at the time of laser treatment as they do for their other cases. This might be done by reporting on the LoSAI scores for all patients.  The LoSAI is the other part of the LoSCAT that the authors cite. The LoSAI scores for erythema, skin thickening, and new or larger lesion (Kelsey, Torok JAAD 2013 PMID: 23562760 ).
3. The authors should comment on whether they considered the induration/skin thickening present at the time of laser treatment represented activity or damage.  
a. By the LoSAI, they would be defined as active if there is skin thickening present
b. However, another morphea study found skin thickening was not specific for activity (Li et al J Rheumatol 2018 PMID: 30219769, DOI: 10.3899/jrheum.171381) so there are likely limitations with the LoSAI.
4. Ultrasound and elasticity measurements
a. The authors should explain more about the ultrasound machine they used, their assessment of reliability and validity of measurements. Accuracy of ultrasound is highly dependent upon the user’s technique and machine (Wortsman J Scleroderma Relat Disord 2019 doi: 10.1177/2397198318799244).
b. Please similarly explain about measurements of elasticity in morphea, as this is not commonly used. There has been a paper describing elastography ultrasound but this is different from what the authors did (Zuccaro et al. UIltrasound Med Biol 2020 PMID: 32951932).  The authors should cite prior work showing their elasticity measure to be helpful for morphea or another similar condition, or else consider stating that this is exploratory.
5. Association between US/elasticity and clinical features.
a. It seems odd that Case 2 had the largest decrease in dermal thickness, but did not have induration, as others have reported induration to be associated with greater dermal thickness (Kreuter et al Arch Dermatol 2005 PMID: 16027298). Could the authors discuss why their findings differ?
b. All the lesions had dermal atrophy, which decreased following treatment. I would expect a decrease in dermal atrophy score to mean there was dermal thickening (less atrophy) but the authors found dermal thinning by ultrasound. How do the authors explain these findings?
c. please explain the pattern of dermal thickness change versus elasticity level.  The cases show differing levels of change for these two values that vary across cases. While Case 2 had the largest decrease in dermal thickness, it had the smallest elasticity change.  Case 1a, 1b, and 3 show a similar level of elasticity change but very different dermal thickness changes.
6. Did the authors do ultrasound and elasticity measurements on contralateral unaffected site? That could provide useful information for evaluating the level of change found to know what is the normal pattern.
a. Dermal thickness and ultrasound pattern varies depending upon anatomic site, age, gender, so comparison between the affected and unaffected side is recommended for interpreting ultrasound findings in morphea (Hoffmann et al Acta Derm 1991. PMID: 1812687).

Author Response

Point 1: LoSDI scores unclear
a. The LoSDI is supposed to be used as a total score of skin damage signs, with the scores of its components summed across all affected anatomic sites, but the authors appear to be reporting scores for specific anatomic regions for case 1. Page 3, lines 133-134, they say that the LoSDI changed from 7 to 3. lines 135-137 discuss expanded laser treatment that includes inguinal lesion, with LoSDI changing from 8 to 4. Authors should clarify whether reported LoSDI are for specific regions and which regions they represent.

Response 1: We used LoSDI as a part of LoSCAT. The LoSCAT assesses 18 cutaneous anatomic sites, capturing both disease activity (mLoSSI) and damage (LoSDI) parameters. Scores for each site are based on the most severe score for each parameter. We used LoSDI for specific anatomic regions.

Point 2:  More clinical features of morphea should be presented.
a. The authors should discuss if induration is present in Case 3 at the time of laser treatment as they do for their other cases. This might be done by reporting on the LoSAI scores for all patients.  The LoSAI is the other part of the LoSCAT that the authors cite. The LoSAI scores for erythema, skin thickening, and new or larger lesion (Kelsey, Torok JAAD 2013 PMID: 23562760 ).

Response 2: Yes, induration was present in Case 3 at the time of laser treatment. We corrected the data.

We decided that due to the possibility of activating the inflammatory process, all laser treatments will be performed in the inactive phase of the disease. LoSAI is the Localized Scleroderma Activity Index. It is based on markers of activity of the disease including erythema, skin induration, and appearance of new/expansion lesions. The assumption of the work was to perform treatments in the inactive phase, so we did not use this scale. We were looking for a scale that would objectively reflect tissue damage. Laser procedures affect only tissue damage, and in no way affect the activity of the disease.

Point 3: The authors should comment on whether they considered the induration/skin thickening present at the time of laser treatment represented activity or damage.  
a. By the LoSAI, they would be defined as active if there is skin thickening present
b. However, another morphea study found skin thickening was not specific for activity (Li et al J Rheumatol 2018 PMID: 30219769, DOI: 10.3899/jrheum.171381) so there are likely limitations with the LoSAI.

Response 3: As mentioned above we performed laser procedures during the inactive phase of the disease. Skin thickening presented in our patients represented tissue damage. We assumed that skin induration is not specific to the activity.

Active lesions are characterized by the lilac ring surrounding the fibrosing center. Over the course of the disease, the sclerotic lesions can become softer again, but can remain indurated. (Kreuter, A., Krieg, T., Worm, M., Wenzel, J., Moinzadeh, P., Kuhn, A., Aberer, E., Scharffetter-Kochanek, K., Horneff, G., Reil, 334 E., Weberschock, T., & Hunzelmann, N. German guidelines for the diagnosis and therapy of localized scleroderma. Journal der 335 Deutschen Dermatologischen Gesellschaft = Journal of the German Society of Dermatology : JDDG 2016. 14(2), 199–216.)

Point 4: Ultrasound and elasticity measurements
a. The authors should explain more about the ultrasound machine they used, their assessment of reliability and validity of measurements. Accuracy of ultrasound is highly dependent upon the user’s technique and machine (Wortsman J Scleroderma Relat Disord 2019 doi: 10.1177/2397198318799244).
b. Please similarly explain about measurements of elasticity in morphea, as this is not commonly used. There has been a paper describing elastography ultrasound but this is different from what the authors did (Zuccaro et al. UIltrasound Med Biol 2020 PMID: 32951932).  The authors should cite prior work showing their elasticity measure to be helpful for morphea or another similar condition, or else consider stating that this is exploratory.

Response 4: Thank you for good advice, we improved the data.

  1. Assessment of reliability and validity of measurements using DarmaLab US were described in this paper Ranosz-Janicka I, Lis-ÅšwiÄ™ty A, Skrzypek-Salamon A, BrzeziÅ„ska-WcisÅ‚o L. An extended high-frequency ultrasound protocol for assessing and quantifying of inflammation and fibrosis in localized scleroderma. Skin Res Technol. 2019;00:1–8. https://doi.org/10.1111/srt.12660
  2. We cite paper which describe conditio with fibrosis: Nguyen, Skin elasticity as a measure of radiation fibrosis: is it reproducible and does it correlate with patient and physician-reported measures? 2014, 13(5), 469–476. https://doi.org/10.7785/tcrtexpress.2013.600257

Point 5: Association between US/elasticity and clinical features.
a. It seems odd that Case 2 had the largest decrease in dermal thickness, but did not have induration, as others have reported induration to be associated with greater dermal thickness (Kreuter et al Arch Dermatol 2005 PMID: 16027298). Could the authors discuss why their findings differ?
b. All the lesions had dermal atrophy, which decreased following treatment. I would expect a decrease in dermal atrophy score to mean there was dermal thickening (less atrophy) but the authors found dermal thinning by ultrasound. How do the authors explain these findings?
c. please explain the pattern of dermal thickness change versus elasticity level.  The cases show differing levels of change for these two values that vary across cases. While Case 2 had the largest decrease in dermal thickness, it had the smallest elasticity change.  Case 1a, 1b, and 3 show a similar level of elasticity change but very different dermal thickness changes.

Response 5: a. We observed that the harder the skin, the slower we achieved the good effect. It also was the youngest of patients and maybe it affected the better result of laser therapy.

  1. Fibrosis can be characterised by increased dermal thickness when we have sclerotic lesions or decreased dermal thickness in atrophic lesions. In our patients, we observe both sclerosis and atrophy. In case atrophy was the dominant feature I would expect dermal thickening, but in this group we didn’t have such case.
  2. A small group of patients is the limitation of the study. It is necessary to srudy a larger and more diverse group of patients. We have started bigger study and additionaly want to compare also histopathological and immunopathological changes in bigger group.

Point 6: Did the authors do ultrasound and elasticity measurements on contralateral unaffected site? That could provide useful information for evaluating the level of change found to know what is the normal pattern.
a. Dermal thickness and ultrasound pattern varies depending upon anatomic site, age, gender, so comparison between the affected and unaffected side is recommended for interpreting ultrasound findings in morphea (Hoffmann et al Acta Derm 1991. PMID: 1812687).

Response 6: We didn’t perform measurements on healthy anatomic areas, but is excellent tip to perform in further research.

Reviewer 2 Report

Dear authors,

Thank you for the opportunity to review this work. I believe it does fill a gap in the literature and in current clinical care, and gives hope to patients with irreversible damage from their disease.

General comments:

The manuscript is too long and wordy. Could remove some of the detail in the introduction and the cases.

Would benefit from review from native English speaker for grammar.

Introduction

- Please select one term morphea or localized scleroderma – and use it consistently.

- The intro could focus on describing the types of irreversible skin damage that can result from morphea and the existing evidence (or lack of) for therapies to correct these.

- Highlight the current gap in research and clinical care. Then clearly state the aim of the study.

Methods

- Indicate the centre /city where patients were seen.

- How were the patients selected?

- Does the patients underlying skin color impact their response to laser treatment?

Results

- It is difficult to say that the improvement is significant. Can you comment on the “minimally clinically significant difference” in the LoSDI score is defined as? Similarly for skin dermal thickness and elsasticity?

Discussion

- Please describe the limitations of a case series, and whether these results can be generalized to other patients.

Author Response

Response to Reviewer 2 Comments

Point 1: The manuscript is too long and wordy. Could remove some of the detail in the introduction and the cases.

Response 1: Thank you kindly for your comments. We described in detail the nature of the disease and its course in our patients due to the rarity of the disease.

Point 2: Would benefit from review from native English speaker for grammar.

Response 2: Thank you, we did it.

Point 3: Please select one term morphea or localized scleroderma – and use it consistently.

Response 3: Both terms mean the same disease. I use them interchangeably to avoid repetition in the text.

Point 4: The intro could focus on describing the types of irreversible skin damage that can result from morphea and the existing evidence (or lack of) for therapies to correct these.

Response 4: I understand this point of view. It is not only a journal for dermatologists, therefore I would like to outline the general nature of the disease in the introduction.

Point 5: Highlight the current gap in research and clinical care. Then clearly state the aim of the study.

Response 5: Thank you, we tried to do this.

Point 6: Indicate the centre /city where patients were seen.

Response 6: It was a group of patients treated in Clinic of Dermatology, Sexually Transmitted Diseases and Clinical Immunology in Olsztyn. We added this information.

Point 7: How were the patients selected?

Response 7: The group consisted of volunteers who felt an aesthetic defect caused by the disease. Patients with active disease phase could not participate in the study.

Point 8: Does the patients underlying skin color impact their response to laser treatment?

Response 8: From our observations, skin color did not affect the results. Our patients had phototype II and III.

Point 9: It is difficult to say that the improvement is significant. Can you comment on the “minimally clinically significant difference” in the LoSDI score is defined as? Similarly for skin dermal thickness and elsasticity?

Response 9:  There is no deifnition as “minimally clinically significant difference” in the LoSDI and in dermal thickness. The evaluation of laser treatments was based on objective scales and measurements with the use of DermaLab. The group of patients is small and we wanted to present the possible good effects of laser treatment. Point10: Please describe the limitations of a case series, and whether these results can be generalized to other patients.

Response 10: A small group of patients is the limitation of the study. It is necessary to study a larger and more diverse group of patients. We have started a bigger study and additionally want to compare also histopathological and immunopathological changes in a bigger group. We added this information.